# A Comparative Analysis of Molecular Biological Methods for the Detection of SARS-CoV-2 and Testing the In Vitro Infectivity of the Virus

**DOI:** 10.3390/microorganisms12010180

**Published:** 2024-01-17

**Authors:** Kalina Shishkova, Bilyana Sirakova, Stoyan Shishkov, Eliya Stoilova, Hristiyan Mladenov, Ivo Sirakov

**Affiliations:** 1Laboratory of Virology, Faculty of Biology, University of Sofia “St. Kl. Ohridski”, 1164 Sofia, Bulgaria; k_shishkova@biofac.uni-sofia.bg (K.S.); sshishkov@biofac.uni-sofia.bg (S.S.); eliya_st@yahoo.com (E.S.); 2Faculty of Dental Medicine, Medical University of Sofia, 1431 Sofia, Bulgaria; biborisova07@gmail.com; 3“AIPPMPDM”, Ltd., 2800 Sandanski, Bulgaria; 4Diagnostic Consulting Center 14, 1408 Sofia, Bulgaria; hristian_m@yahoo.com; 5Department of Medical Microbiology, Medical Faculty, Medical University of Sofia, 1431 Sofia, Bulgaria

**Keywords:** nested PCR, LAMP, nasal swabs, SARS-CoV-2, diagnostic, IVDR

## Abstract

The virus discovered in 2019 in the city of Wuhan, China, which was later identified as SARS-CoV-2 and which spread to the level of a pandemic, put diagnostic methods to the test. Early in the pandemic, we developed a nested PCR assay for the detection of SARS-CoV-2, which we validated and applied to detect the virus in feline samples. The present study describes the application of the nested PCR test in parallel with LAMP for the detection of the virus in 427 nasopharyngeal and oropharyngeal human samples taken between October 2020 and January 2022. Of the swabs tested, there were 43 positives, accounting for 10.1% of all samples tested, with the negatives numbering 382, i.e., 89.5%, and there were 2 (0.4%) invalid ones. The nPCR results confirmed those obtained by using LAMP, with results concordant in both methods. Nasal swabs tested using nPCR confirmed the results of oropharyngeal and nasopharyngeal swab samples tested using LAMP and nPCR. The focus of the discussion is on the two techniques: the actual practical application of the laboratory-developed assays and the diagnostic value of nasal samples. The nPCR used is a reliable and sensitive technique for the detection of SARS-CoV-2 in nasopharyngeal, oropharyngeal, and nasal swab samples. However, it has some disadvantages related to the duration of the entire process, as well as a risk of contamination. Experiments were performed to demonstrate the infectivity of the virus from the positive isolates in vitro. A discrepancy was reported between direct and indirect methods of testing the virus and accounting for its ability to cause infection in vitro.

## 1. Introduction

The first representatives of Coronaviridae were isolated in the early 1930s. These were the causative agents of infectious bronchitis in chickens [1] and transmissible gastroenteritis in pigs [2]. Since then, research on coronaviruses has been carried out mainly because of the serious economic losses they cause in animal husbandry and because they are a suitable model for the study of viral pathogenesis. The focus of attention shifted with the emergence of SARS-CoV-1 in 2002 [3] and MERS-CoV in 2012 [4]. These events prepared the scientific community (to some extent) for the 2019 outbreak of the respiratory disease in Wuhan. The etiological agent was shown to have 79–82% homology to SASR-CoV and 50% to MERS-CoV. It was classified in the same genus, *Betacoronavirus*, subfamily *Orthocoronavirinae*, and was named SARS-CoV-2 [5,6,7,8]. The disease brought about challenges not only in terms of treatment but also in terms of diagnostic techniques and approaches.

SARS-CoV-2 mainly affects the respiratory and gastrointestinal tract [9]. The symptoms include high fever, cough, myalgia, arthralgia, and fatigue, headache, shortness of breath, nasal discharge, diarrhea, hemoptysis, and loss of smell and taste [10,11,12,13]. The diagnosis of SARS-CoV-2 is based on the clinical picture, radiological imaging [14], X-ray images captured using deep learning approaches [15], serological tests [16,17,18], and virus detection using antigen and molecular techniques [19,20]. Routine SARS-CoV-2 diagnosis and detection in humans use IVD real-time RT PCR kits [21,22], LAMP [23], and rapid tests [24]. Various variants of nested PCRs [25,26,27,28] have also been developed, mainly for scientific purposes rather than as IVD human diagnostic tests. Many researchers have conducted comparative analyses between different methods for the detection of SARS-CoV-2, e.g., comparisons between various rapid antigen tests [24,25,29], between antigen tests and real-time RT PCR [30,31], and between various real-time RT PCR tests [29,32,33], LAMP, and PCR [34].

The clinical samples for diagnosis include nasopharyngeal (NP) swabs, oropharyngeal (OP) swabs, saliva [31,35,36], throat swabs, bronchoalveolar lavage, tracheal aspirate, and sputum [34,37,38]. NP sampling, if performed improperly, can cause various complications [39] such as the breaking of the swab stick and bleeding [40,41]. In addition, respiratory viruses are also released through exhaled air and serous discharge when replicating in the respiratory tract [42,43], hence the retention of viral particles in the nose (when the virus is not replicating there) due to the protective function of the mucosa and nasal hair [44]. On the other hand, some studies suggest that nasal swabs may have a lower sensitivity for SARS-CoV-2 detection than NP swabs [39,45,46,47]; a combined OP/NS approach has been found to match NP performance [46]. This is most probably why NP and OP swabs were established in diagnostic practice in Bulgaria, but not nasal swab samples [48,49,50].

Our previous study validated and applied nested (n) One-Step RT PCR for SARS-CoV-2 detection in swab samples [51] and sera from cats [52]. The assay showed high sensitivity, as it was able to detect low concentrations of virus RNA. This prompted us to apply nOne-Step RT PCR for SARS-CoV-2 detection in human nasopharyngeal, oropharyngeal, and nasal swab samples, in parallel with the routine IVD LAMP assay, as well as to assess the diagnostic value of nasal swabs in the detection of SARS-CoV-2 infection.

## 2. Materials and Methods

### 2.1. Samples

The present study included nasopharyngeal and oropharyngeal swab samples collected as part of the process for diagnosing SARS-CoV-2 infection. The samples were taken from 427 persons who visited Diagnostic and Consultative Center 14 (Sofia, Bulgaria) for testing for SARS-CoV-2 infection. Individuals aged from 2 to 89 years were studied, their average age being 41.68 ± 20.30 years. Samples were taken from persons with clinical symptoms/temperature, contact persons, and persons leaving the country, as well as persons wishing to be examined. Samples were collected according to the protocol. When nasal diagnostic specimens were examined, samples were taken from both nostrils of each individual with separate swabs, and the swabs were placed together in an Eppendorf tube. The collected oropharyngeal and nasopharyngeal samples were placed in Eppendorf tubes containing an antibiotic, Gentamicin 40 mg/1 mL (Sopharma, Bulgaria), in 500 µL saline solution. The samples were taken between October 2020 and January 2022.

All patients or their legal guardians completed diagnostic test request forms and signed informed consent forms giving permission for the samples to be used for SARS-CoV-2 testing. The number of samples taken in each year of the study period varied: 45 patients approached the diagnostic center for examination in 2020, 341 were examined in 2021, and 41 tests were performed in 2022.

### 2.2. Loop-Mediated Isothermal Amplification (LAMP)

For routine detection of SARS-CoV-2 in this study, the SARS-CoV-2 RT-LAMP kit (Vienna BioCenter, Vienna, Austria) and the LAMP IVD test Ender Mass (Switzerland) were used, following the manufacturers’ instructions for RNA extraction and subsequent reactions—40 µL of the lysis buffer was put into a 1.5 mL reaction tube with 100 µL of the sample. The mixture was heated for 2 min at 95 °C. The reaction volume was 50 µL, and the reaction mixtures contained 30 µL of enzyme, 8 µL PCR water, 4 µL of primer mix, and 8 µL of the prepared sample. The following program was used: 30 min at 65 °C, collecting fluorescence data once per minute. A temperature gradient from 80 °C to 90 °C was applied for the assessment of the melting temperature of the amplification product, and the fluorescence was continuously recorded.

### 2.3. RNA Extraction and Nested One-Step RT PCR

Prior to nOne-Step RT PCR, virus RNA was extracted (from all samples tested using LAMP) using the viral RNA/DNA extraction kit (Jena Bioscience, Jena, Germany) according to the manufacturer’s instructions.

Detection of SARS-CoV-2 RNA in the samples was performed using the One-Step RT PCR kit (Qiagen, Germantown, MD, USA). The primers were constructed on the basis of the N gene positions in the SARS-CoV-2 genome according to sequence MN908947.3, isolate Wuhan-Hu-1 [53]. External primers: Ext2019nCorV F 5′-GGCAGTAACCAGAATGGAGA-3′ (positions 28,346–28,365) and Ext2019nCorV R 5′-CTCAGTTGCAACCCATATGAT-3′ (positions 28,681–28,661) defining a 335 bp fragment; and internal primers: intF 5′-CACCGCTCTCACTCAACAT-3′ (positions 28,432–28,450) and intR 5′-CATAGGGAAGTCCAGCTTCT-3′ (positions 28,643–28,624) defining a 212 bp fragment. The reaction volume was 25 µL, and the reaction mixtures contained 5 µL of RNA, 1 µL of dNTPs, 1 µL of Enzyme mix, 5 µL of RT buffer, 11 µL of PCR water, and 2 µL of primers. The following program was used: 50 °C—30 min for reverse transcription and 95 °C—15 min for enzyme activation, followed by 40 cycles of denaturation at 95 °C—20 s, annealing at 54.6 °C—25 s and elongation at 72 °C—40 s; and final extension at 72 °C—10 min and store at 10 °C. The second round of reaction was carried out in 25 µL, and the reaction mixtures contained 2 µL of cDNA, 12.5 µL of Master mix (Bioline, Meridian, MS, USA), 8.5 µL of PCR water, and 2 µL of primers intF-R. The following program was used: 30 cycles of denaturation at 95 °C—10 s, annealing at 54.6 °C—20 s, and elongation at 72 °C—30 s; and final extension at 72 °C—10 min and store at 10 °C. SaCycler-96 (Sacace Biotechnologies, Como, Italy) and FluoroCycler (Hain Lifescience, Nehren, Germany) were used.

We also applied the method in one reaction as a conventional One-Step RT PCR in the manner described above [51], but using only the inner primers and reducing the elongation temperature from 40 to 30 s, due to the smaller fragment size expected with these primers.

### 2.4. Quantity and Quality Control of Extracted RNA and PCR Products

The quantity and quality control of extracted RNA and PCR products was conducted using a DNA/RNA calculator (Pharmacia LKB, Cambridge, UK) and gel electrophoresis. Gel electrophoresis was performed in 2% agarose (Lonza, Walkersville, MD, USA), 10 ng/mL ethidium bromide (Sigma, Livonia, MI, USA), 100 mL 1 × TAE buffer, and 1-kb DNA ladder (Bioline, London, UK) at 120–150 V, 70–120 mA for 10–40 min. Visualization was performed using a UV transilluminator at 240/260 nm. When measuring the resulting RNA and cDNA using the DNA/RNA calculator, the sample was diluted 1:30 or 2 µL of sample and 58 µL of distilled water. When conducting gel electrophoresis after the PCR of the obtained product, we used 10 µL.

### 2.5. Cultivation of the Virus in Cell Culture

The VERO E6 cell line gene clone 76 (obtained from ATCC CRL-1587) was cultured in Dulbecco’s Modified Eagle Medium (DMEM) (Sigma-Aldrich, Burlington, MA, USA, Merck, Darmstadt, Germany) with low glucose, 20 mM Hepes buffer (Sigma-Aldrich, Merck, Germany). Furthermore, 10% fetal calf serum (FCS) (Sigma) was added to the medium to culture the cells and 4% to prepare a maintenance medium. In addition, 96-well plates were used. Cells were seeded and, upon reaching a dense cell monolayer, inoculated with 100 microliters of undiluted virus suspension. This was followed by incubation in a thermostat at 37 °C for 1 h to adsorb the virus. After the adsorption time had elapsed, the appropriate volume of support medium was added, again 100 microliters. Incubation in a thermostat at 37 °C followed, and virus-induced morphological changes in the cells—the cytopathic effect (CPE)—were monitored daily.

### 2.6. Statistical Analysis

Data analysis was performed using analysis of variance, descriptive statistics, and the chi-square test. The established dependencies were also determined using the Cramer’s coefficient V = 0.359, which is a statistically significant coefficient with a significance level of Approx. Sig. = 0.000 < α = 0.05.

## 3. Results

The study was carried out in the period October 2020–January 2022 and included 427 individuals: 201 males (47.1%) and 226 females (52.9%). The gender distribution is shown in Figure 1.

The results of the chi-square test showed that the gender distribution depended on the year of sampling (Pearson’s chi-square = 43.152, significance level Asymp. Sig. (two-tailed) = 0.000 < α = 0.05) (Figure 1b). The test subjects ranged in age from 2 to 89 years, with a mean age of 41.68 ± 20.30 years. For 50% of the sample, the mean age was 43 years, and the predominant group was that of Mode = 54 years. It is noteworthy that in 2020 and 2022, the number of samples provided by women predominated.

### 3.1. Loop-Mediated Isothermal Amplification (LAMP) and Nested One-Step RT PCR

The results of the conducted experiments are shown in Figure 2. Bands with sizes of 212 bp were registered.

Figure 3 shows the invalid results obtained in the examination of two of the samples.

After measuring the purity of the obtained RNA using a DNA/RNA calculator (Pharmacia LKB, UK) for nOne-Step RT PCR, the samples showed low values—ratio 0.34 and 0.51.

The PCR analysis showed 44 positive results (10.1% of all tested individuals), 381 negative ones (89.5% of all tested individuals), and 2 invalid tests accounting for 0.4% of all tested individuals (Figure 4).

Positive samples were repeated with the one-step variant of the nPCR method, and the results were confirmed, with bands in nine of the samples having a slightly lower luminescence intensity. The applied nPCR method showed little difference in sensitivity between outer and inner primers [51]. In this regard, the lower luminescence intensity of nine of the samples when tested with the one-step variant of the method may be due to this difference, which, in the presence of less viral nucleic acid in the starting sample, may lead to a worse visual reaction/lighting.

The distribution of positive tests by sampling method was as follows: 18 oropharyngeal swabs, or 40.5% of the positive test group and 4.0% overall relative proportion; 25 nasopharyngeal secretions, or 57.1% of positive tests and a total relative share of 5.7%; and oropharyngeal and nasopharyngeal secretions in one person, or 2.4% of the group of positive tests and an overall relative proportion of 0.2%. Examinations of oropharyngeal samples were made on 216 persons, which represents a relative share of 50.6%, and nasopharyngeal samples were taken from 209 persons and examined, which is a relative share of 48.9% of the total number of people examined. Both types of samples were obtained from two people and examined, which represents a relative share of 0.5% of the total number of 427 people (Table 1). A nasal swab was taken from the oropharyngeal or/and nasopharyngeal areas of all the subjects and examined in the manner described in the Material and Methods section; this represents 100% of the subjects, which is why it is not included in the analysis presented in Table 1.

The results from the nOne-Step RT PCR test of the nasal swab samples confirmed those from the NP and OP swabs obtained using both LAMP and nOne-Step RT PCR. Therefore, there is a 100% match between the results of both nOne-Step RT PCR and LAMP tests.

### 3.2. Cultivation of the Virus in Cell Culture

Given the fact that the detection of a viral genome or viral proteins does not always signal the presence of infectious virions after testing by direct detection of viral genome, we also investigated using the methods of conventional virology the infectivity of the virus and its ability to replicate in cell cultures. After establishing the number of individuals according to the need for testing and the positive samples among them, presented in Table 2, we cultivated the isolates in cell cultures according to the methodology described in the Materials and Methods section.

All SARS-CoV-2 positive samples were cultured. The aim of this study was to determine whether the presence of a viral genome indicates the presence of infectious virions capable of replication. The results showed that, of the patients who requested a test and tested positive, 24 (89%) showed the presence of an infectious virus. In the patients with clinical symptoms and the contact ones, the positive samples also proved to be positive when examined using classic virological methods, namely, establishing the proof of an infectious virus. Patients with clinical symptoms and contacts who showed positive molecular biological detection tests also showed the presence of an infectious virus. Of those who took a detection test and wished to leave the country, four of the five positive samples showed the presence of an infectious virus.

The results of culturing the virus and the cytopathic effect shown are presented in Figure 5.

The first manifestations of a cytopathic effect were observed on the first day after monolayer inoculation of the virus. Microscopically, fields of clustered cells with altered morphology (reduced size and jagged contours) were detected. Two days after virus inoculation into cell culture, all cells in the monolayer were infected, and the cytopathic effect unfolded, being observable throughout the cell monolayer. The cytopathic effect was expressed in shrinking of the cells (a reduction in their diameter), a loss of adhesive ability, and detachment of the cells from the bottom of the culture vessel. Some of the cells were lysed. Our team cultivated human coronavirus 229E in MDBK cell culture. We found that it was successfully cultured in this cell line [54]. This virus belongs to the genus alpha coronavirus. In contrast, in our attempts to cultivate SARS-CoV-2, which belongs to the Genus *Betacoronavirus*, in the same cell line, it became apparent that there was no cytopathic effect at 48 h.

## 4. Discussion

The total number of samples from the two genders differed non-significantly for the whole period of the study, 2020–2022 (Figure 1a). The analysis of the gender distribution within each year showed a significant difference in 2021, when the number of tested samples was the largest (Figure 1b). It could be speculated that the male predominance in the number of samples tested in 2021 might, at least in part, be attributable to the lack of lockdown in the country and, consequently, to the (physical) return to work of more men than women, with the associated SARS-CoV-2 testing and issuing of certificates [54,55]. Such a suggestion is supported by the descriptive statistics and the age distribution histogram, in which the tested individuals were predominantly in the active working-age population, with a median age of 43 years and mode (predominant age) of 54 years (Figure 2). Also, a number of factors (such as imposed restrictions and social media) had an impact at that stage of the pandemic on the motivation of people to get tested [56], especially those with a suspected infection following contact with a sick person or one exhibiting a clinical manifestation of infection. These considerations could also serve as an explanation for the low percentage of positive samples.

The results from this study confirmed the high sensitivity of the nOne-Step RT PCR assay for the detection of SARS-CoV-2 [51] and other conditions [25,26,27,28]. We found that the nPCR results coincided with those from the reference method when applied to human clinical samples. This can be explained by the fact that the primers are virus-specific, regardless of the type of sample, and the nucleic acid is purified, not just extracted. As the tests are designed to detect the same virus but in different hosts, the host species (human) could affect the results when the extraction technique is specifically developed for the subsequent reactions and is an inseparable part of the kit, such as the LAMP assay [51]. With this in mind, we chose a universal, column-based method for virus RNA/DNA extraction, which allows the use of the obtained NAs in various subsequent assays, such as those able to be performed using most kits on the market.

The applied nOne-Step RT PCR method showed little difference in sensitivity between outer and inner primers [51]. In this regard, the lower luminescence intensity of nine of the samples when tested with the one-step variant of the method may be due to this difference, which, in the presence of less viral NC in the starting sample, may lead to a more poor visual response/glow.

Although the results from the two methods coincided, the nested One-Step RT PCR procedure has its purely technical disadvantages, such as a risk of contamination and a considerably longer time of execution [34,55,57]. Application of the one-step variant of the method shortens the time required to receive a result to 2 h and 40 min (including RNA extraction and depending on the number of samples). It also reduces the risk of contamination, which is a significant risk when performing nPCR [51].

We can note as an advantage of the LAMP method used the fact that it eliminates the subjective factor in the reading of a color reaction, but it requires a single-channel real-time PCR apparatus, which has a higher market price than a conventional PCR apparatus; this, on the other hand, can be regarded as a disadvantage of this kind of LAMP.

The nOne-Step RT PCR assay was theoretically developed at the beginning of 2020, but normative restrictions (REGULATION (EU) 2017/746) prevented its routine application in SARS-CoV-2 diagnostic practice. With the introduction of the IVDR Regulation (EU) 2017/746), the problems in this regard may become deeper [56,58], as in any future pandemics, laboratories (research and private ones) would not be able to respond independently, promptly, and adequately to the situation. This has fueled criticism against lobbyism in the new EU legislation for allowing only private companies to perform such activities in the EU [56,58], thus, allegedly, paving the way for the privatization of knowledge and scientific research. Concerns have been voiced that with this legislation, along with the proposed international agreement on pandemic prevention and preparedness [57,59], any efforts for laboratories to develop methods such as this one and others will become futile [25,26,27,28,58,59,60,61], as will their actual application in helping people to adequately control diseases, thus making the EU dependent on corporations and the WHO.

The samples that produced invalid results in both tests indicate that the sample quality may have been compromised, e.g., during the sampling step, during the NA extraction procedure, or during the detection assays [60,61,62,63,64]. We believe that, because each batch of the LAMP diagnostic tests is validated against a positive clinical sample, it is unlikely that the invalid results are attributable to a failure of the tests, as suggested by Matzkies et al. [63,65]. Owing to the invalid results, the same patients gave a second sample the same day, but the results from these swabs were not included in the statistical analysis.

The lower diagnostic accuracy with nasal samples compared to nasopharyngeal ones in previous reports [39,45,46,47] may be due to a combination of factors, such as the low concentration of viral particles in the nose [64,66] and the diagnostic limitations of the methods. nOne-Step RT PCR is a highly sensitive molecular technique for NA detection in various types of samples [25,27,65,67]; therefore, we applied this assay for SARS-CoV-2 detection in nasal samples. The results obtained from the nasal swabs in this study overlapped with the results from the other types of samples (nasopharyngeal and oropharyngeal swabs) tested using LAMP and nOne-Step RT PCR. This could be due to a high concentration of viral particles in the nose (depending on the stage of the infectious process) and/or the sensitivity of the method, which is 0.015 ng/μL [51]. The complications that may occur when collecting swab samples from the nasopharynx [39,40,41] may affect the test results [66,68]. On the other hand, our results, together with the fact that the OP/NS combination is comparable to NP, which is defined as a gold standard [46,67,69], suggest that it may be possible to reconsider the types of samples needed to detect SARS-CoV-2. From a practical point of view, we consider the use of methods from conventional virology to test and isolate an infectious virus to be a valuable contribution to the development of the science of virology. The fact that this particular virus, in contrast to the human coronavirus 229E, does not show a clear cytopathic effect in the MDBK cell line points to further study of the relevant receptors necessary for productive infection. A significant number of genetic differences between Vero E6 sublines [70] have been identified, including single nucleotide variants, indels, and copy number variations. Sublineage-specific enriched loss-of-function and missense variants have been identified that potentially contribute to differences in response to viral infection among Vero sublines. Variants of ACE2, which functions as a receptor for SARS-CoV, were found to be heterozygous in Vero JCRB0111, Vero CCL-81 and Vero 76; However, Vero E6 contains only an isoleucine allele, resulting from the loss of one of the X chromosomes. This research provides a new field for the study of receptors and penetration of SARS-CoV. We believe that the study and proof by cell culture biological methods of the infectivity of a part of the isolates, which were confirmed positive by the application of molecular biological methods, adds additional information to the improvement of the methods of testing the ability of the virus to cause a productive infection. We consider it an advantage to culture SARS-CoV-2 virus isolated from patients, given that the literature on the subject is limited and demonstration of viral infectivity is of great importance. We consider particularly important the fact that not all positive samples when applying molecular biological methods are positive for the presence of an infectious virus. The reasons for this could be various, including a small number of infectious units that are not sufficient to cause infection. This fact is particularly significant when restricting the movement of people.

## 5. Conclusions

The Nested One-Step RT PCR assay used is this study was reliable and sensitive for the detection of SARS-CoV-2 in nasopharyngeal, oropharyngeal, and nasal swabs. The drawbacks that the method has are associated with the time-consuming preparation step (DNA extraction) and the procedure itself, as well as the risk of contamination. The one-step variant of the method shortens the procedure time and reduces the risk of contamination. In the context of this and other studies [45,68,71], we also recommend the use of nasal swabs for SARS-CoV-2 detection. It is perhaps of value to think in the direction of developing new tests based on viral infectivity, since there is a discrepancy between the results obtained in detection by indirect and direct methods in respect of people limited by certificates for whom the ability to travel is important.

## Figures and Tables

**Figure 1 microorganisms-12-00180-f001:**
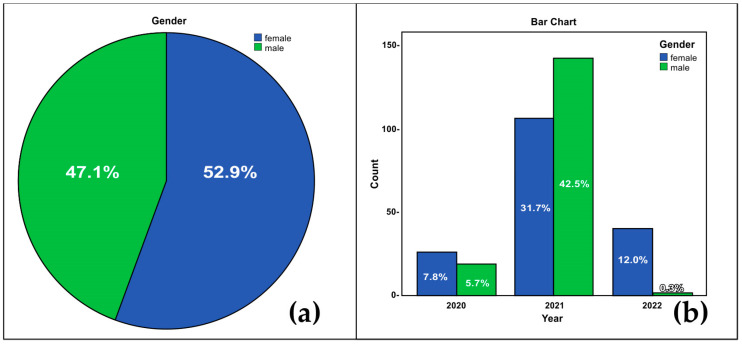
Gender distribution of individuals tested for SARS-CoV-2, descriptive statistics (**a**); and distribution in each year of sampling, chi-square test (**b**).

**Figure 2 microorganisms-12-00180-f002:**
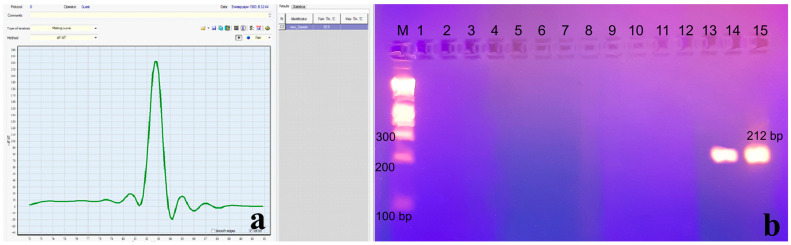
Nasal swab results of samples tested using LAMP (**a**) and conventional nOne-Step RT PCR (**b**) with the internal primer pair yielding a 211 bp fragment. (**a**) Sample 84; (**b**) M—DNK Ladder 100 bp (Bioline, Meridian, MI, USA), 1—negative control, 2–13—negative samples from 84 to 95; 14—positive sample 83, 15—positive control.

**Figure 3 microorganisms-12-00180-f003:**
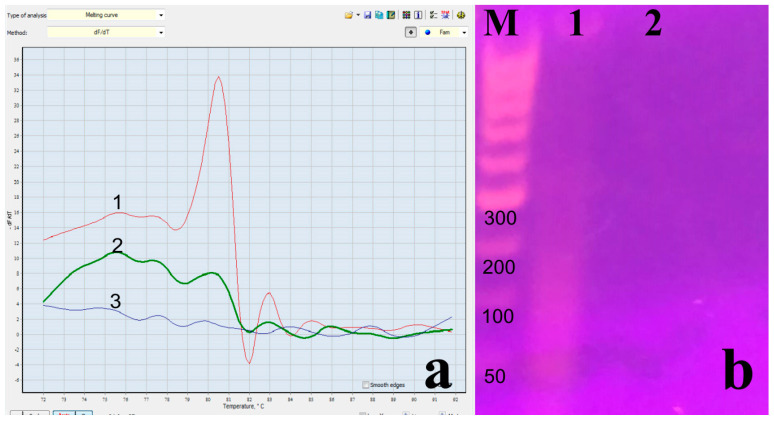
Invalid result of samples tested using (**a**) LAMP Ender Mass test: 1—Control, 2—Sample 420, 3—Sample 421; (**b**) Negative control: Sample 421 tested by nOne-Step RT PCR—Lane 1, and Lane M—DNA Ladder.

**Figure 4 microorganisms-12-00180-f004:**
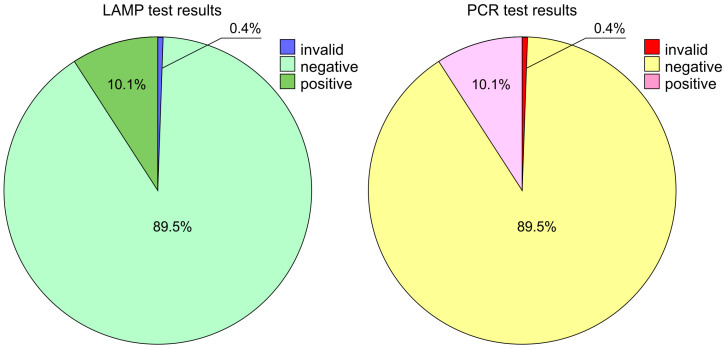
Analysis of variance of categorical variables for distribution of LAMP and nOne-Step RT PCR test results.

**Figure 5 microorganisms-12-00180-f005:**
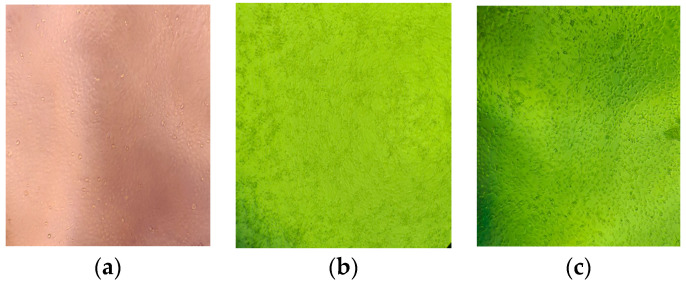
Cell control (**a**); virus-infected cells from an isolate—24th h (**b**); virus-infected cells from an isolate—48th h (**c**).

**Table 1 microorganisms-12-00180-t001:** Cross-tabulation of bivariate frequency distributions—part of chi-square, nOne-Step RT PCR, and LAMP test results according to the sampling method.

Oropharyngeal/Nasopharyngeal Result nOne-Step RT PCR Test Result Cross-Tabulation						
			nOne-Step RT PCR Test Result			Total
			Invalid	Neg	Pos	
Oropharyngeal/ Nasopharyngeal results	1-Oropharyngeal	Count	0	198	18	216
		% within OP/NP result	0.0%	92.1%	7.9%	100.0%
		% within PCR test result	0.0%	52.0%	40.5%	50.6%
		% of total	0.0%	46.6%	4.0%	50.6%
	2-Nasopharyngeal	Count	2	182	25	209
		% within OP/NP result	1.0%	87.4%	11.6%	100.0%
		% within PCR test result	100.0%	47.8%	57.1%	48.9%
		% of total	0.5%	42.8%	5.7%	48.9%
	1 + 2	Count	0	1	1	2
		% within OP/NP result	0.0%	50.0%	50.0%	100.0%
		% within PCR test result	0.0%	0.3%	2.4%	0.5%
		% of total	0.0%	0.2%	0.2%	0.5%
Total		Count	2	381	44	427
		% within OP/NP result	0.5%	89.6%	10.1%	100.0%
		% within PCR test result	100.0%	100.0%	100.0%	100.0%
		% of total	0.5%	89.6%	10.1%	100.0%

Note: OP—oropharyngeal; NP—nasopharyngeal..

**Table 2 microorganisms-12-00180-t002:** Distribution of persons according to the need for testing.

Need for Test					
		Frequency	Percent	Valid Percent	Cumulative Percent
Valid	By own will	27	6.3	79.4	79.4
	Clinical symptoms—fever	1	0.2	2.9	82.4
	contact with positive person	1	0.2	2.9	85.3
	traveling out of the country	5	1.2	14.7	100
	Total	34	8	100	
Missing	N/A	393	92		
Total		427	100		

## Data Availability

The data presented in this study can be found in the manuscript.

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
