# Peer review of "A Comparative Analysis of Molecular Biological Methods for the Detection of SARS-CoV-2 and Testing the In Vitro Infectivity of the Virus"

_microorganisms, 2024, doi:10.3390/microorganisms12010180_

Round 1

Reviewer 1 Report

Comments and Suggestions for Authors

There is a need for a better presentation of the results, a greater description of them, enriching the figure and table captions, and explaining their content.

Sensitivity and specificity data are missing for the analyzed methods, compared with the cytopathic effect in Vero Cells due to the infection.

How do these results behave depending on the year of the samples where they were isolated? Low sensitivity or specificity?

Reviewer 2 Report

Comments and Suggestions for Authors

Revision manuscript “Comparative analysis of molecular biological methods for detection of SARS – CoV-2 and proving the in vitro infectivity of the virus”

In this Communication, the authors compare two different molecular methods (LAMP, nOne Step Rt PCR) on nasopharyngeal and oropharyngeal swabs obtaining the same results. In addition, nasal swabs tested with nested PCR confirmed the results of oropharyngeal and nasopharyngeal swab samples tested with LAMP and nPCR. The authors conclude that nested PCR is an effective method for detecting SARS-CoV-2, although increases the risk of contamination, and it is time consuming. Furthermore, nasal swab could be a valuable sample for detecting SARS-CoV-2.

The authors stress also the importance of proving the infectivity of a patient by inoculating the sample positive by molecular method in VERO E6 cells.

I understand the concerns of the authors regarding the impossibility of using the nested PCR in SARS-COV-2 diagnostics because of the restrictions imposed by the EU regulations. However, only an automated and certified workflow can guarantee reliable results.

Figure 2: a high quality picture showing the negative and positive samples is required. Allow a longer migration of the bands in order to distinguish clearly the different fragment sizes. An arrow outside the gel could be used to indicate the 211 bp fragment size. Furthermore, a lower amount of DNA could be loaded into the wells since the bands look very thick.

In the text, the authors do not mention the two invalid results shown in Figure 3.

Table 1: check the percentage value (50.6%) reported in the column Total, third line. In addition, the total count reported in the Table is 423, should not it be 427?

Minor points:

-Lines 120 and 124, why did you store the samples a 10°C instead of 4°C as usual?

-Line 216, (52) should be in black.

-Lines 338 and 342: maybe “cellular biological methods” rather than “molecular biological methods” could be used in this case.

Comments on the Quality of English Language

English can be improved.
